# Molecular Features and Treatment Paradigms of Acute Myeloid Leukemia

**DOI:** 10.3390/biomedicines12081768

**Published:** 2024-08-06

**Authors:** Mihir Shukla, Maher Abdul-Hay, Jun H. Choi

**Affiliations:** Department of Hematology and Medical Oncology, NYU Langone Health, Perlmutter Cancer Center, New York, NY 10016, USA; mihir.shukla@nyulangone.org (M.S.);

**Keywords:** acute myeloid leukemia, targeted therapy, molecular

## Abstract

Acute myeloid leukemia (AML) is a common hematologic malignancy that is considered to be a disease of aging, and traditionally has been treated with induction chemotherapy, followed by consolidation chemotherapy and/or allogenic hematopoietic stem cell transplantation. More recently, with the use of next-generation sequencing and access to molecular information, targeted molecular approaches to the treatment of AML have been adopted. Molecular targeting is gaining prominence, as AML mostly afflicts the elderly population, who often cannot tolerate traditional chemotherapy. Understanding molecular changes at the gene level is also important for accurate disease classification, risk stratification, and prognosis, allowing for more personalized medicine. Some mutations are well studied and have an established gene-specific therapy, including FLT3 and IDH1/2, while others are being investigated in clinical trials. However, data on most known mutations in AML are still minimal and therapeutic studies are in pre-clinical stages, highlighting the importance of further research and elucidation of the pathophysiology involving these genes. In this review, we aim to highlight the key molecular alterations and chromosomal changes that characterize AML, with a focus on pathophysiology, presently available treatment approaches, and future therapeutic options.

## 1. Introduction

Acute myeloid leukemia (AML) is a heterogeneous cluster of myeloid neoplasms that remains a challenging disease to treat. It is a disease of the elderly with a median age of around 67 years. Despite rigorous research in the past decades, the prognosis remains poor, with a 5-year relative survival (OS) of 15–25% [1]. The initial steps in leukemic transformation involve the accumulation of genetic aberrations, including somatic mutations or epigenetic changes in immature cells of myeloid origin and/or abnormal changes in chromosomes, leading to the activation of pathways that promote the aberrant clonal proliferation of affected cells [2]. Accordingly, AML is risk-stratified by recurrent cytogenetic and molecular markers such as FMS-like tyrosine kinase (FLT)-3, nucleophosmin-1 (NPM-1), and other mutations. Assessing molecular characteristics is ever more important, as new diagnosis, classification, and management guidelines have been updated, most strictly utilizing genetic abnormalities rather than morphological definitions [3,4]. The prognosis of AML differs dramatically according to the molecular stratification: complete remission with chemotherapy is achievable in around 60–85% of patients with favorable-risk AML while only about 15–55% of patients achieve remission in the poor-risk group [5,6,7]. Therefore, the genomic screening of patients with AML is highly useful for deciding upon an effective treatment for these patients.

Induction regimens are largely divided into high-intensity chemotherapy and less intensive chemotherapy. Intensive chemotherapy is built around a backbone of traditional cytotoxic drugs such as cytarabine and anthracycline given at high doses, with the purpose of eradicating leukemic blast cells to an undetectable level. The most common regimen is the “7 + 3” treatment using cytarabine and anthracycline, which have been the standard therapy for AML for several decades [5]. In the past two decades, a series of new research studies have created effective low-intensity therapy regimens designed for elderly populations or patients with comorbidities unfit for intense chemotherapy. The most well-known example of low-intensity therapy is the regimen combining a hypomethylating agent (HMA) and venetoclax (VEN), which produced a near equivalent response rate to traditional intensive chemotherapy [8]. The explosion of research in high-throughput DNA and RNA sequencing technology has revealed the extensive genetic changes that occur in AML. The implementation of this research and the development of targeted therapy have further refined and improved these standard regimens, allowing a more personalized approach in treating patients with AML. Currently, 40–50 genes that harbor recurrent somatic mutations in various AML subtypes have been discovered, with an average of about 7 to 13 mutations per case. The evidence suggests that at least five genes are recurrently mutated and known as “driver” mutations, while the rest of the mutations are random and defined as “passenger mutations” [9,10,11]. Only a small proportion of somatic mutations have a targeted therapy developed against them, including FLT3 inhibitors and IDH1/2 inhibitors. In this review, we will provide an overview of genomic alterations in AML and highlight the important pathophysiology, its prognostic significance, available therapies, and active future research.

## 2. Somatic Mutations in AML

### 2.1. FLT3

*FMS*-like tyrosine kinase 3 (FLT3), which is a part of the class III receptor tyrosine kinase receptor group, plays a pivotal role in hematopoiesis [12]. Several studies have characterized the nature of FLT3 expression in human and murine cells and identified its importance in early hematopoiesis (CD34+cells), and also shown that FLT3 expression can be high in hematologic malignancies such as acute myeloid leukemia (AML) [12]. The FLT3 ligand is also expressed in hematopoietic cells, and is thought to contribute to the immune response via the stimulation of early B-cells, dendritic cells, and natural killer cells; it is thought that this immune response may predict an anti-tumor effect [12]. However, the FLT3 ligand (also referred to as ligand for flt3/flk2 receptor) has been shown to stimulate leukemic blasts, particularly in concert with other cytokines (G-CSF, GM-CSF, IL3, SCF) [12,13]. Thus, the role of FLT3 and the FLT3 ligand in immunity and leukemia is complex, with features suggestive of an anti-tumor response but also leukemic stimulation. It should also be noted that mutated FLT3-ITD responds to the FLT3 ligand, and that leukemia cells express the FLT3 ligand; this has important implications when considering conditions that may increase the FLT3 ligand, such as the post-chemotherapy plasma milieu [14].

The normal function of the FLT3 receptor includes activation via phosphorylation, followed by downstream intracellular signaling, leading to a proliferative response; mutated FLT3 remains constitutively activated [15]. The main FLT3 mutations are internal tandem duplication (ITD), which is a gain-of-function mutation to the receptor tyrosine kinase, and the tyrosine kinase domain (TKD), which causes a loss of auto-inhibition [16]. The TKD mutation is a point mutation in the kinase domain activation loop [14].

In patients with AML, the FLT3-ITD mutation can be seen in about 20–25% of adult patients [14,15,16,17], while TKD mutations are found in about 7% [14]. Mutations in FLT3-ITD have historically been associated with a poor prognosis [15,17]. FLT3-ITD, in particular, has been described as being associated with a poor overall survival (OS) and early relapse [16]. The influence of the FLT3-TKD mutation on the prognosis is less clear. A retrospective analysis of 3082 AML patients found that FLT3-TKD had no significant influence on OS or event-free survival (EFS) [18], while a meta-analysis found that both mutations have a negative prognostic impact on disease-free survival (DFS). This study, however, was limited by the analysis of the observational studies and abstracted data, heterogeneity, and publication bias [19]. Another meta-analysis concluded that FLT3-TKD did not have a significant effect on DFS or OS (pooled HR of DFS 1.12 and OS 0.98) [20]. The 2022 ELN risk classification by genetics at initial diagnosis includes mutated *NPM1* with FLT3-ITD and wild-type *NPM1* with FLT3-ITD (without adverse-risk genetic lesions) in the intermediate risk category, while mutated NPM1 without FLT3-ITD is included in the favorable risk category, regardless of the FLT3 allele ratio [21]. In the 2017 ELN risk stratification, the allelic ratio of FLT3-ITD had implications for risk; for example, wild-type *NPM1* with FLT3-ITD-high (allelic ratio ≥ 0.5) was classified as adverse, whereas wild-type NPM1 without FLT3-ITD or with FLT3-ITD-low (allelic ratio < 0.5) without adverse genetic lesions had an intermediate risk [22]. This nuance regarding the FLT3-ITD allelic ratio is no longer in the 2022 ELN risk classification by genetics at initial diagnosis [21].

Various FLT3 inhibitors have been extensively studied and are widely used in the treatment of AML today. The multi-kinase inhibitor sorafenib was the first-generation TKI assessed in a phase 1 study in patients with refractory or relapsed AML, which found that 10% of patients (five patients with an FLT3-ITD mutation) had complete remissions (CRs) or complete remissions with incomplete count recovery (CRi) [23]. In the phase 2 setting, the SORAML trial evaluated the addition of sorafenib versus placebo to standard induction and consolidation chemotherapy in patients with newly diagnosed AML, and found that the median EFS in the placebo group was 9 months, versus 21 months in the sorafenib group [24]. A recently published phase 2 study evaluated induction chemotherapy with either sorafenib or placebo in patients with new AML with the FLT-ITD mutation; sorafenib did not improve event-free survival (EFS), and there was no significant difference in two-year OS between sorafenib versus placebo [25]. More data are needed for sorafenib in the phase 3 setting. Midostaurin, which is a multitargeted kinase inhibitor, has been studied in a phase 3 trial in patients with newly diagnosed AML with FLT3 mutations in the RAFITY trial [26]. In the RATIFY trial, patients of ages 18 and 59 were assigned standard intensive chemotherapy induction and consolidation with either the addition of midostaurin or placebo; the OS was significantly increased in the midostaurin versus the placebo group (HR for death 0.78, *p* = 0.009) [26]. The authors conclude that although the study was not powered for subgroup analysis, the OS was significantly longer with midostuarin in the FLT3-TKD, FLT3-ITD-high, and FLT-3 ITD-low groups [26]. Midostaurin is now approved to be used in conjunction with intensive induction chemotherapy followed by consolidation [27]. Other TKIs have been studied, mostly in the relapsed/refractory setting. The ADMIRAL study was a phase 3 study that evaluated gilteritinib versus salvage chemotherapy in the FLT3-mutated relapsed/refractory AML setting. Gilteritinib demonstrated a significantly longer median OS compared to chemotherapy (9.3 months versus 5.6 months), with higher percentages of remission [28]. Gilteritinib is approved by the Food and Drug Administration (FDA) for adults with R/R FLT3-mutated AML [29]. In the induction/consolidation and maintenance phases of treatment for newly diagnosed, non-favorable risk AML, a phase 1b study evaluated gilteritinib in combination with induction/consolidation and maintenance treatment. A total of 36 of the 58 participants had *FLT3*-mutated AML, and those who had FLT-mutated AML achieved a composite complete response of 89%; however, there were concerns about delayed count recovery, and a longer count recovery time was associated with a high gilteritinib trough concentration [30]. In *FLT3*-mutated AML patients not eligible for intensive chemotherapy, a phase 3 trial evaluated gilteritinib with AZA versus AZA alone; there was a statistically significant difference in composite complete remission rates between gilteritinib/AZA (58.1%) versus AZA alone (26.5%), though it failed to meet the primary objective of a statistically significant median OS difference [31]. The combination of gilteritinib with AZA and VEN in both the frontline and relapsed/refractory setting was studied in a phase I/II trial for *FLT3*-mutated AML, and it was found that the CR/CRi rate was 96% in the frontline cohort and 27% in the relapsed/refractory setting [32]. Quizartinib is another second-generation inhibitor that solely targets FLT3-ITD that has also been compared to salvage chemotherapy in relapse/refractory FLT3-ITD AML in the phase 3 QuANTUM-R study. Quizartinib significantly improved the median OS (mOS) compared to chemotherapy (6.2 months versus 4.7 months) [33]. The QuANTUM-First phase 3 trial studied quizartinib combined with induction and consolidation chemotherapy, versus placebo combined with induction and consolidation chemotherapy, in newly diagnosed FLT-ITD-positive AML. This study found a median OS of 31.9 months in the quizartinib group versus 15.1 months in the placebo group (HR 0.78, *p* = 0.032) [34]. This combination is now approved by the FDA [35]. Crenolanib was evaluated in a phase II study among patients with FLT3-mutated AML in the relapsed/refractory setting (median 3.5 prior therapies); at a median follow up of 14 weeks, the ORR was 47%, with a 12% CRi [36]. Another study evaluating crenolanib in the refractory/relapsed AML with FLT3-ITD setting found that crenolanib resulted in a 39% CRi and 11% partial response (PR) in those patients who had not received FLT3 inhibition previously, with an OS of 234 days [37]. In the group that had prior TKI exposure, the OS was 94 days [37]. Phase 2 trials examining crenolanib in combination with chemotherapy have also been conducted, and phase 3 trials comparing crenolanib as a maintenance or salvage therapy are ongoing (NCT03250338, NCT03258931) [16].

FLT3 inhibitors have also been studied in the maintenance setting after allogeneic hematopoietic stem cell transplantation (HSCT). In a phase 1 trial, sorafenib was used as a maintenance therapy in patients with FLT3-ITD AML (started between D45 and D120 after HSCT), resulting in a 1-year progression-free survival (PFS) of 85% and 1-year OS of 95% [38]. The phase 2 SORMAIN study evaluated 83 patients with FLT3-ITD-positive AML after HSCT and demonstrated that the hazard ratio (HR) for relapse or death was 0.39 in the sorafenib versus placebo group [39]. Another study by Xuan et al. assessed sorafenib in an open-label, phase 3 setting, where patients with FLT3-ITD AML were assigned sorafenib maintenance versus non-maintenance at 30–60 days after transplantation. The results showed a decrease in the 1-year cumulative incidence of relapse (CIR) in the sorafenib (7%) versus control group (24.5%) [40]. Midostaurin has also been evaluated in the phase 2 RADIUS trial (open-label) in patients with FLT3-ITD AML after allogeneic HSCT. This trial compared the standard of care with or without midostaurin, and found an estimate 18-month RFS of 89% in the midostaurin arm compared to 76% in the standard of care only arm [41]. Gilteritinib as a post-allogenic HSCT maintenance was being studied in the phase 3 BMT CTN Protocol 1506 [42]. In this study (NCT02997202), adult patients with FLT3-ITD AML who were treated with HSCT in first remission were assigned to either gilteritinib or placebo for 24 months in the post-HCST maintenance setting; the primary endpoint of RFS was not found to have a statistically significant difference [43]. However, patients with measurable residual disease (MRD) treated with gilteritinib had an improved RFS and OS compared with placebo; importantly, this study highlights the rationale for use of MRD in this setting [43]. This idea, regarding the clinical utility of MRD evaluation in FLT3-ITD AML, builds on recent studies which have highlighted the prognostic value of MRD testing in FLT3-ITD AML [44,45]. Quizartinib has been studied as a maintenance after allogenic HSCT in a phase 1 study in patients with FLT3-ITD-mutated AML; four patients (31%) stopped quizartinib due to adverse toxicity (NCT01468467) [46]. Table 1 highlights the targeted therapies that are approved by the FDA for FLT3 (and also IDH1/2, as discussed in a later section).

Despite the promise of FLT3 inhibition as a targeted molecular approach to AML treatment, resistance mechanisms have been identified that may affect the efficacy of this therapeutic approach. Several primary and secondary resistance patterns to FLT3 inhibition have been identified, and thus combination strategies to overcome such resistance mechanisms are being investigated [47].

It is clear that FLT3 inhibitors have been extensively studied and shown to have clinical utility in FLT3-ITD mutated AML. The recent emergence of *FLT3* MRD monitoring as a prognostic tool may significantly affect treatment planning and decisions, further highlighting the importance of molecular testing in AML.

### 2.2. NPMI Mutation

The *nucleophosmin* (*NPM1*) gene is vital to cellular function, and *NPM1* mutation is present in roughly 30% of adult AML patients [48]. Indeed, in AML with normal cytogenetics, *NPM1* is commonly mutated and is considered to be the most frequent gene mutation [49]. The nucleophosmin protein moves between the nucleus and cytoplasm, and has several cellular functions including tumor suppressor regulation, DNA repair, and transcription, and stress response [50]. With a clear role in tumorigenesis, aberrant NPM expression is noted in both solid tumors and hematologic malignancies [50]. The wild-type nucleophosmin protein resides in the nucleolus; however, mutated *NPM1* leads to abnormal protein localization, and ultimately results in the protein in the cytoplasm; this displacement, and an altered homeobox (*HOX*) gene expression, may contribute to leukemogenesis in *NPM1* mutants [49,51].

The World Health Organization (WHO) 5th edition classifies AML with mutated *NPM1* as a distinct entity, despite the blast count, perhaps due to the rapid time to progression from myelodysplastic syndrome (MDS) or myelodysplastic syndrome/myeloproliferative neoplasm (MDS/MPN) [52]. It confers a favorable prognosis, with an OS rate of about 40% and CR rate of 80% with traditional intensive induction chemotherapy [21]. Frequent co-occurring mutations include *FLT3*, *DNMT2A*, and *IDH1/2* [9]. The treatment of AML with the *NPM1* mutation traditionally includes intensive induction chemotherapy or low-intensive chemotherapy with HMA and VEN, followed by consolidation and/or allogeneic stem cell transplantation, although patients who achieve MRD negative CR can attain long-term survival without allo-HSCT. Combining traditional chemotherapy with novel targeted agents such as FLT3 inhibitors and VEN is being explored, and more specific targeted approaches are being studied [53].

There is a theoretical basis for targeting exportin 1 (XPO1), which functions in the nuclear export of proteins, including tumor suppressor proteins, and is thought to contribute to the cytoplasmic exportation of mutated *NPM1* via increased nuclear export signals in mutant *NPM1* [53,54,55]. Indeed, it is thought that the inhibition of XPO1 and the consequent return of mutated NPM1 to the nucleus can lead to the differentiation of AML cells [49]. XPO1 inhibitors have been studied in human AML. In a phase 1 study, Selinexor, a selective inhibitor of nuclear export compound (SINE) that functions to block XPO1, was studied in a dose escalation trial in patients with relapsed or refractory AML, adults > 60 years old with MDS or unfavorable cytogenetics, and patients > 70 years old who were not chemotherapy candidates; 17% of the patients enrolled had *NPM1* mutations [56]. Of the 81 patients that were considered to be evaluable, 5 patients had a CR and 2 patients had a CRi, with 45 patients developing stable disease (SD); in the entire cohort, the median PFS was 1.7 months and median OS was 2.7 months [56]. Another phase 1 trial examined the use of Selinexor and decitabine in adult patients with relapsed/refractory AML or in elderly, untreated, unfit AML patients; of the 25 patients included, 10 had a response, and the median OS for the cohort was 5.9 months with a PFS of 5.9 months, though several adverse effects of treatment were noted [57]. In responding patients, the PFS was 11.8 months and the OS was 12.9 months, compared to PFS 4.4 months and OS 5.9 months in non-responding patients [57]. Selinexor has also been studied in combination with high-dose cytarabine and mitoxantrone in adult patients with newly diagnosed AML or with relapsed/refractory AML, in a phase 1 study, which included five (25%) patients with *NPM1* mutations [58]. The authors reported an ORR of 70% [58]. In a phase 2 study of 42 adult patients with relapsed/refractory AML, Selinexor was studied with cytarabine and idarubicin, with an overall response rate (ORR) of 47.6%, with median OS 8.2 months, representing a median OS of 12.6 months in the cohort receiving Selinexor 40 mg/m^2^ twice weekly (for 4 weeks) versus a median OS of 8 months in the cohort receiving an absolute dose of 60 mg twice weekly (3 weeks out of a 4-week cycle), which was not a statistically significant difference [59]. This study had four patients with *NPM1* mutation, with three of these patients responding with CR [59]. Another phase 2 study evaluated Selinexor compared to physician’s choice in adult, elderly AML patients with R/R AML, and showed no statistically significant difference in the median OS; treatment-emergent adverse events occurred in all Selinexor-treated patients and 95.2% of the physician choice cohort [60]. Given the toxicity profile of Selinexor, second-generation SINE options are being studied, such as KPT-8602 (eltanexor), which is shown to be active in patient-derived xenograft models of AML blasts, and additionally was shown to be better tolerated in mice compared with Selinexor [61]. This improved tolerability may be due to the decreased crossing of the blood–brain barrier of KPT-8602 compared to Selinexor [55,61]. Preclinical studies suggest that sustained XPO1 inhibition, such as can be attained via eltanexor, leads to down-regulation of HOX/MEIS and may exert an improved anti-leukemic effect, though clinical data are needed to evaluate this comprehensively in humans [51].

Another modality of treatment in AML that may also have a mechanistic role in NPM1-mutated AML is targeting B-cell lymphoma 2 (BCL2) proteins [53]. BCL2 proteins are crucial components of the mitochondrial apoptotic response, and high expression is seen in AML cells; BCL2 inhibition via VEN (VEN) has been shown to improve outcomes in elderly patients with AML when combined with hypomethylating agents (HMAs) [8]. AML patients with NPM1 mutations have shown a composite remission of 66.7% in the phase 3 trial of VEN plus azacytidine (AZA), and the CR/CRi/CR with partial hematologic recovery was 78% in the trial with LDAC plus VEN [8,62]. With intensive chemotherapy plus VEN, NPM1-mutated AML patients achieved an 80–100% CR/CRi rate [63,64]. A retrospective study by Lachowiez et al. also showed that patients of an age > 65 and NPM1-mutated AML treated with HMA/VEN had a higher OS compared to those treated with HMA monotherapy or intensive chemotherapy [65].

Other molecular-targeting drugs in NPM1-mutated AML include menin inhibitors and spleen tyrosine kinase (SYK) signaling, which have been reviewed in Ranieri et al. [53]. Particularly, NPM1-mutated AML cells are dependent on menin–*MLL* interaction for leukemic cell differentiation via HOX1 and MEIS1 expression, making them susceptible to menin inhibition [66]. The details of menin inhibition will be discussed in the KMT2A section.

Immunotherapy may present another potential treatment strategy in *NPM1*-mutated AML. Mutated *NPM1* AML may have characteristic peptides that can be targeted via T-cells with specific T-cell receptors (TCRs) that identify such peptides [67]. Though clinical research will be needed to substantiate the utility of immunotherapy in treating *NPM1*-mutant AML, in vitro models, and a mice model, have shown that CAR-T cell therapies may have anti-leukemic utility while mitigating off-tumor toxicity [68].

*NPM1* mutations can be detected qualitatively and quantitatively, by molecular techniques, immunohistochemistry, and flow cytometry [69]. An *NPM1* mutation minimal residual disease (MRD) assessment via an RNA-based RQ-PCR has shown utility for predicting the remission duration and relapse risk [70]. Molecular monitoring may identify a potential relapse prior to morphologic relapse and therefore offer a window for pre-emptive treatment, prior to progression from molecular to morphologic relapse [71]. Emerging data suggest that molecular MRD testing for *NPM1*-mutant AML may help guide therapy after remission, particularly by using MRD status to aid transplantation decisions [72]. The monitoring of the *NPM1* mutation via qPCR in the post-transplant setting has also been shown to have utility [73].

Figure 1, which was adapted and re-created from the cited articles and associated figures in those articles, describes the complex interactions of wild-type *NPM1* and mutated *NPM1* (NPM1c), and the cellular consequences of NPM1c [48,74,75].

### 2.3. IDH

In 2009, Mardis et al. published a report of data from DNA sequencing to characterize the genome of cytogenetically normal AML and noted the presence of a mutation in the isocitrate dehydrogenase (*IDH*) 1 gene [76]. Mutations in *IDH* are thought to cause the formation of 2-hydroxyglutarate (2HG, or D-2-hydroxyglutarate (D-2-HG)), an oncometabolite, via a reduction of alpha-ketoglutarate, which has also been described in gliomas; this is shown in Figure 1 [77,78,79,80,81]. 2HG causes DNA hypermethylation and thus alters hematopoietic differentiation [82]. *IDH1*/2 mutations are present in about 20% of AML patients [77,83]. The in vitro data suggested that the selective inhibition of IDH2 via small molecule AGI-6780 induced the differentiation of AML cells and TF-1 erythroleukemia cells [83,84]. Additionally, AG-221, an oral selective inhibitor of *IDH2* mutant enzymes, demonstrated 2HG suppression in preclinical models [83]. In a first-in-human phase 1/2 trial examining enasidenib (AG-221/CC-90007) in *IDH2*-mutated relapsed or refractory AML, the ORR was 40.3% and the median OS was 9.3 months [85]. Additionally, enasidenib has been shown to induce a response in older adults with *IDH2*-mutant AML, with an ORR of 30.8% [86]. Enasidenib in combination with azacitadine, compared with azacitadine alone, has also been studied in a phase 1b/2 trial in adults with newly diagnosed IDH2-mutant AML; in the phase 2 portion, there was an ORR of 74% for patients in the combination group versus 36% for patients in the azacitadine monotherapy group [87]. IDH1 has also been demonstrated as a therapeutic target; in a phase 1 study assessing the use of ivosidenib in *IDH1*-mutated AML, the rate of CR or CR with partial hematologic recovery was 30.4%, and the ORR was 41.6% [88]. In newly diagnosed AML patients who were not deemed to be eligible for standard therapy, the rate of CR and CR with partial hematologic recovery was 42.4%, and the median OS was 12.6 months [89]. The combination of either ivosidenib or enasidenib with induction/consolidation chemotherapy in newly diagnosed IDH1/2-mutated AML has been studied in a phase 1 trial that showed induction CR rates of 55% for ivosidenib and 47% for enasidenib [90]. The combination of ivosidenib in addition to AZA was studied in a phase 1b trial in patients with newly diagnosed *IDH1*-mutated AML who were ineligible for induction chemotherapy, and it found an ORR of 78.3% and a CR rate of 60.9% [91]. Ivosidenib has also been evaluated in a phase 3 trial in combination with AZA versus AZA monotherapy in patients with newly diagnosed *IDH*-1-mutated AML; the median OS was 24 months in the combination AZA/ivosidenib group, versus 7.9 months in the AZA/placebo group [92].

Currently, ivosidenib is approved as a monotherapy or with azacitidine for the treatment of IDH1-mutated newly diagnosed AML in adults aged 75 years or older or those ineligible for intensive induction chemotherapy, as well as a monotherapy to treat adult patients with relapsed or refractory AML [93,94,95]. It is worth noting that a phase 3 trial comparing enasidenib to conventional care in older patients with R/R AML and the *IDH2* mutation failed to meet its primary endpoint of an improved OS, leading to its market withdrawal in Canada [96,97].

Olutasidenib (FT-2102) is a selective IDH1 inhibitor that was recently given FDA approval for treatment in R/R AML [98,99,100]. Phase 1 results in patients with *IDH*-mutant AML or MDS (R/R or treatment-naïve patients were included) had shown encouraging response rates [99,101,102]. In a phase 2 study evaluating olutasidenib in patients with R/R AML with the *IDH* R132 mutation who had not previously received IDH1 inhibitors, the CR/CRh rate was 35% and the ORR was 48%, with notable adverse events of febrile neutropenia and cytopenia [99,103].

Both ivosidenib and enasidenib have been studied in the post-allogeneic hematopoietic stem cell transplant (HSCT) setting as a maintenance treatment. In a phase 1 trial in patients with *IDH1*-mutated AML, ivosidenib was initiated 30–90 days following HSCT; the two-year CIR of relapse was 19%, and non-relapse mortality was 0%, with two-year PFS 81% and two-year OS 88% [104]. In patients with IDH2 myeloid malignancies, enasidenib was studied as a maintenance after HSCT; the CIR of relapse was 16%, and the two-year PFS was 69% and OS was 74% [105].

Table 1 highlights the FDA-approved targeted therapies for *IDH1/2* mutations (in addition to *FLT3*, as mentioned above). There are ongoing clinical trials assessing the role of IDH inhibition in combination with other modalities, such as in combination with induction/consolidation (NCT03839771), or in combination with CPX-351 in relapsed AML (NCT03825796) [106]. Several other trials are active, examining the role of IDH-targeted therapies in AML (NCT02677922, NCT03515512, NCT02719574, NCT02632708).
biomedicines-12-01768-t001_Table 1Table 1FDA-approved targeted therapies for FLT3, IDH1, and IDH2 mutations.DrugMechanismSettingMajor TrialMidostaurinMulti-kinase inhibitorNewly diagnosed *FLT3*-mutated AML (in combination with chemotherapy) [27]RATIFY [26]Gilteritinib Highly selective, multi-kinase inhibitorR/R *FLT3*-mutated AML [29]ADMIRAL [28]Quizartinib Selective kinase inhibitorNewly diagnosed *FLT3*-ITD-mutated AML (in combination with chemotherapy in induction/consolidation) and as maintenance monotherapy following consolidation [35]QuANTUM-First [34]EnasidenibIDH2 inhibitorRelapsed or Refractory AML with *IDH2* mutation [107]AG221-C-001 [85]IvosidenibIDH1 inhibitorNewly diagnosed AML (in elderly or unable to receive intensive induction) with susceptible *IDH1* mutation (alone or in combination with azacitadine), relapsed or refractory AML with susceptible *IDH1* mutation [93,94,95]AG120-C-001 [88,89], AG120-C-009 [92] Olutasidenib IDH1 inhibitorRelapsed or refractory AML with susceptible *IDH1*
mutation [100]Study 2102-HEM-101 [102,103]

### 2.4. TP53

Present in less than 10% of AML patients, alterations in *TP53*, a well-established tumor suppressor gene encoding p53, portend a poor prognosis and chemoresistance [108]. The ELN 2022 classification identifies AML with mutated *TP53* as a distinct classification, associated with complex karyotypes and a poor prognosis [21]. Indeed, the median survival in *TP53*-mutated AML is about 5–10 months [109]. In patients with newly diagnosed AML treated with 10-day decitabine and VEN, one study found that *TP53* mutations conferred a reduced ORR compared to wild-type *TP53* (66% versus 89%, *p* = 0.002), with a significantly higher 60-day mortality in the *TP53*-mutated group (26% versus 4%, *p* < 0.001); the OS was significantly worse in *TP53*-mutated patients compared to wild-type *TP53* at 5.2 months versus 19.4 months, respectively [110].

In a first-in-human study, p53 was targeted via APR-246, which restores the transcriptional function of mutant p53 in patients with hematologic malignances and prostate cancer; biologic effects such as cell cycle arrest and early signals of apoptosis were observed, and one patient with p53-mutated AML in this study demonstrated a reduced blast percentage [111]. In the phase 2 trial evaluating eprenetapopt (APR-246) in combination with AZA in patients with *TP53*-mutated MDS and AML, AML patients demonstrated an ORR of 33%, and the median OS in AML patients with less than 30% marrow blasts was 13.9 months and 3 months in AML patients with more than 30% marrow blasts [112]. Eprenetapopt was also studied with AZA in the post-HSCT maintenance setting in *TP53*-mutant AML and MDS in a phase II trial (NCT03931291) with a median OS of 20.6 months [113]. Another trial, in the phase Ib/II setting, evaluated eprenetapopt with AZA in *TP53*-mutated myeloid disorders, including MDS and AML, and reported ORR 64% and CR 36% in AML patients [114].

Magrolimab is a humanized IgG4 monoclonal antibody to CD47; by binding CD47, which is an anti-phagocytic protein, magrolimab enhances the macrocytic phagocytosis of cancer cells [109]. Phase 1b data from a trial that assessed magrolimab with AZA for frontline treatment in patients with untreated AML ineligible for intensive chemotherapy revealed a CR of 31.9% and median OS of 9.8 months in the *TP53*-mutant group [115]. However, the phase 3 ENHANCE-2 trial which sought to evaluate frontline magrolimab and AZA was discontinued, due to an analysis suggesting no survival benefit compared to SOC, and the phase 3 ENHANCE-2 trial which sought to evaluate magrolimab in combination with VEN plus showed an increased risk of death, and it is currently not recruiting patients (NCT04778397) [116,117].

Interleukin-3 receptor alpha (IL-3Ra) is thought to be overexpressed in patients with AML, and the expression of IL-3Ra is thought to correlate with a greater number of blasts and a poorer prognosis [118]. CD123, an antigen for IL3, is expressed on AML cells and thus has emerged as a potential therapeutic target [109,119]. Via a dual-affinity retargeting (DART) molecule from CD3 and CD123, AML cells can be targeted for treatment [119]. Flotetuzumab, a bispecific DART to CD3e and CD123, has been studied in the context of *TP53* mutations in AML; a post hoc analysis of patients with *TP53*-altered R/R AML treated with flotetuzumab showed a CR in 47% of patients [120]. It has been proposed that certain immune environment signatures in AML, particularly the “immune-infiltrated” profile, predicts resistance to chemotherapy but a potential response to immunotherapeutic options such as flotetuzumab [121,122]. In a phase 1/2 study evaluating the use of flotetuzumab in adults with R/R AML, those patients with primary induction failure or early relapse had an ORR (CR/CRh/CRi) of 30%; the median OS was 10.2 months in those who had CR/CRh [122].

Though not the most common gene alteration in AML, *TP53* alterations predict a poor prognosis and chemotherapy refractoriness. Notably, the addition of VEN to the hypomethylating agent often does not provide a significant benefit to the HMA alone in treating *TP53*-mutant AML [110]. Therefore, further research will be needed to treat *TP53*-mutated AML patients, including immunotherapeutic options as above.

### 2.5. KMT2A/MLL

The mixed-lineage leukemia (*MLL*) gene is involved in the regulation and maintenance of HOX expression [123]. Thus, translocations in *MLL* can lead to a dysregulation of *HOX* expression and consequently drive leukemia [123]. In the case of *MLL* regulation and its relationship to oncogenesis, it is thought that both wild-type *MLL* and aberrantly translocated *MLL* contribute to HOX regulation [124]. *MLL* is located on chromosome 11q23, and the products of *MLL* fusion are thought to convert HSC to leukemic cells. *MLL* translocations are seen in AML, acute lymphoblastic leukemia, treatment-related leukemia, and bi-phenotypic leukemia [125].

The *MLL* gene is also referred to as the lysine methyltransferase 2A (KMT2A) gene, and generally mutations in this gene are associated with a poor prognosis [126]. The presence of chromosomal rearrangements in the *MLL* gene, also noted as 11q23/KMT2A, has implications for prognosis. A retrospective study found that t(9;11) had an intermediate prognostic significance, while t(6;11), t(11;19), unbalanced 11q23, and t(11;v)(q23;v) were predictive of a poor prognosis [127]. Another study found that young patients with t(9;11)(p22;q23)/KMT2A-*MLL*T3 fared better than other intermediate risk groups, as designated by the 2017 ELN risk classification, while elderly patients with the same translocation had outcomes similar to adverse ELN patients [128]. The 2022 ELN risk classification stratifies t(9;11)(p21.3;q23.3)/*MLL*T3::KMT2A AML in the intermediate risk category, and t(v;11q23.3)/KMT2A-rearranged AML in the adverse risk category [21].

The function of the *MLL* gene is thought to be associated with a tumor suppressor protein called menin (from the *MEN1* tumor suppressor gene), which is crucial to the leukemogenic potential of mutated *MLL* as it relates to HOX expression [129,130]. It is also thought that menin inhibition may have a therapeutic effect in NPM1-mutated AML due to its effect on HOX and MEIS1 expression [66]. This is described further in Figure 1 [75].

A phase 1 trial examined the role of revumenib, an oral menin–KMT2A inhibitor, in R/R acute leukemia, and found a 30% rate of CR/CRh [126]. Revumenib is also being evaluated in the phase 2 setting for patients with R/R KMT2A-rearranged acute leukemia (AUGMENT-101); it should be noted that this study includes patients with acute lymphoblastic leukemia/mixed-phenotype acute leukemia [131]. The interim data provide the results of 94 patients who were heavily pretreated, including prior allo-HSCT in 50% of patients (83% AML), and who received ≥1 dose of revumenib. Grade ≥ 3 adverse events were observed in 54.3% of patients—commonly, differentiation syndrome (16%), febrile neutropenia (13.8%), and QTc prolongation (13.8%). A total of 22.8% achieved CR + CRh, of whom 70% achieved MRD negativity, and the CRc was 43.9%, with a duration of response of 6.4 months [131]. Another study, KOMET-001, is a phase 1/2 first-in-human trial investigating Ziftomenib, a KMT2A–menin inhibitor, in R/R AML; the cohort is composed of about 33% KMT2A-rearranged patients and 13% NPM1-mutated patients (the phase 2 portion is planned to assess NPM1-mutated AML) [132]. Multiple trials investigating menin inhibitors to treat AML are recruiting patients, either as monotherapy (NCT04811560) or in combination with other chemotherapy (NCT05326516, NCT06001788, NCT05453903, NCT04988555, NCT05886049, NCT06284486, NCT04752163). In the frontline setting, SNDX-5613 is being investigated with intensive chemotherapy in new AML (NCT06226571), and ziftomenib is being evaluated in conjunction with VEN/AZA, VEN, or 7 + 3 induction (NCT05735184). JNJ-75276617 is an oral inhibitor of KMT2A–menin binding that may have activity in treating KMT2A or NPM1 altered AML [133]. JNJ-75276617 is being studied in a phase 1b trial in patients with AML with NPM1 or KMT2A alterations (NCT05453903). Pinometostat (EPZ-5676) is a histone methyltransferase disrupter of telomeric silencing 1-like (DOT1L) inhibitor that showed early promise in a phase 1 study in advanced leukemia, including patients with *MLL*-r (NCT01684150) [134]. Pinometostat is also being studied in the phase Ib/II setting in patients with R/R *MLL*-r AML or patients deferring induction treatment with *MLL*-r AML, in combination with AZA [135].

More recently, it has been shown that IKAROS degradation via mezigdomide has demonstrated pre-clinical activity in KMT2A-r and NPM1-mutated AML cells, and may also have efficacy in combination with menin inhibition [136].

Given the prognostic significance of KMT2A/*MLL* re-arrangement, further research into therapeutic drug development targeting will be vital to improving outcomes in patients with AML harboring this alteration. Menin inhibitors have so far shown promising results, and future FDA approval is imminent.
Figure 1This figure demonstrates the effects of key mutations and the effect on cellular function. In the cytoplasm, isocitrate is converted to alpha-ketoglutarate (A-KG), but IDH1 mutations lead to the reduction of A-KG to D-2-hydroxyglutarate (D-2-HG) which is an oncometabolite that travels to the nucleus and inhibits TET2, which blocks DNA demethylation; additionally, D-2-HG is created via reduction in the mitochondria by *IDH2* mutant enzymes from Krebs cycle-generated A-KG [77,78,79,80,81]. IDH1 inhibitors target the cytoplasmic reduction of A-KG to D-2-HG, while IDH2 inhibitors target the same but in the mitochondria [79,80,81]. NPM1, which normally resides in the nucleolus and minimally binds XPO1, can travel to the nucleoplasm in conditions of stress, where it inhibits HDM2; the inhibition of HDM2 is significant because the normal function of HDM2 is to inhibit *TP53* [48,74]. Thus, by inhibiting HDM2, NPM1 can increase *TP53* which has important implications for cell regulation in stressful conditions [48,74]. Mutant NPM1 (NPM1c) has a higher affinity to XPO1 and thus is prone to nuclear export, which leads to the export of important nuclear proteins [48,74]. Additionally, the consequent result of mutant *NPM1*, and XPO1-NPM1c, can lead to increased HOX expression [48,74]. Additionally, NPM1c and KMT2Ar interact with menin, which facilitates leukemogenic cellular changes; this can be targeted via menin inhibition [75]. This figure was adapted from the figures and text in the sources that are cited in this section.
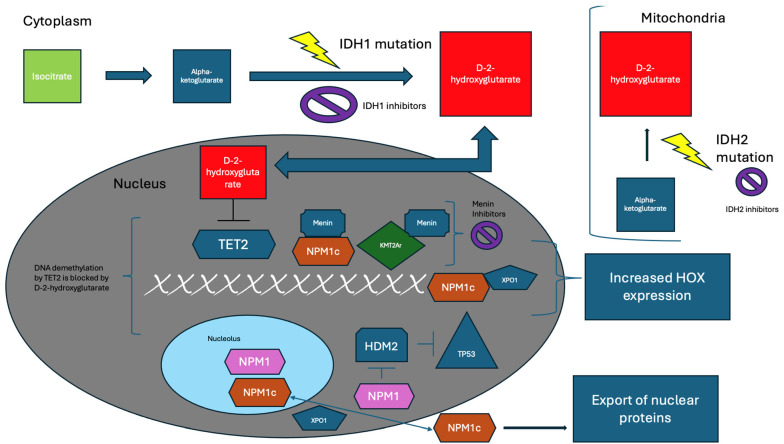


### 2.6. EZH2

The lysine methyltransferase called enhancer of zeste homolog 2 (EZH2) is a component of the polycomb repressive complex 2 (PRC2), and it plays a complex, multifaceted role in gene expression for stem cell differentiation; mutations in EZH2 are seen in 4% of patients with AML [137]. EZH2, via its catalysis of lysine 27 on histone H3 (H3K27me3), plays a vital role in gene expression, and EZH2 overexpression is traditionally associated with cancer progression; interestingly and uniquely, AML induced by *EZH2* alteration is thought to be related to loss of function in the context of *EZH2* as a tumor suppressor, whereas *EZH2* may demonstrate oncogenicity in the maintenance setting [137,138]. An in vitro murine model demonstrated that there may be therapeutic utility in EZH2 inhibition (via GSK343 which is an S-adenosyl methionine-competitive inhibitor) in AML; EZH2 inhibition tested on primary human leukemia cells also demonstrated growth inhibition [138]. Additionally, in an in vivo mouse model of AML1-ETO9a secondary leukemia treated with EPZ-6438 EZH2 inhibition, survival was prolonged with EZH2 inhibition compared to control [138]. Given the reported poor prognosis of *EZH2*-mutated AML patients, targets for treatment may have important prognostic implications [138,139].

The inhibition of EZH2 with tazemetostat has been studied in a phase 2 clinical trial in patients with relapsed/refractory follicular lymphoma; this study showed an 69% objective response rate in the EZH2-mutated cohort [140]. Tazemetostat has also been studied in other solid tumors and lymphoid malignancies [141]. However, the data for EZH2 inhibition in myeloid malignancies such as AML seem to be less robust. Mechanistically, it has been suggested, via ex vivo and in vivo experiments in AML cells and patient samples, that EZH2 inhibition (GSK126) may participate in the chromatin decondensing of AML blasts via decreased H3K27me3, and subsequently this may increase chemosensitivity when used in conjunction with standard chemotherapy [142].

### 2.7. PHF6

The plant homeodomain finger 6 gene (*PHF6*) is a tumor suppressor gene known to be involved in T-cell acute lymphoblastic leukemia (T-ALL) (20%) but that has also been noted in about 3% of adult AML in one study [143]. Patel et al., in a mutational analysis, found PHF6 in 3% of cases, and noted that *PHF6* was associated with a lower OS [144]. It is thought that PHF6 protein plays a role in the regulation of transcription [145]. In a mouse model examining the function of PHF6 knockout in *MLL:AF9* fusion AML cells, it was found that *PHF6* knockout negatively affected leukemic proliferation, with a more pronounced effect at later disease stage [146]. One study noted a negative prognostic significance of high *PHF6* expression [147]. While *PHF6* is thought to play a tumor suppression role in T-ALL, its role in AML may be pro-oncogenic; one study in mouse models found that *PHF6* overexpression may contribute to leukomogenesis, and *PHF6* deletion may negatively affect AML development [148]. Hou et al. propose that *PHF6* may indirectly increase BCL2 expression via the binding of p50, thus leading to the inhibition of apoptosis via the nuclear factor kappa B (NF-κB) pathway, thus identifying NF-κB inhibition as a potential therapeutic target [148].

### 2.8. U2AF1

U2 small nuclear RNA auxiliary factor 1 (U2AF1) is an integral part of the splicing of pre-mRNA, and mutations in *U2AF1* are more commonly seen in MDS, but can also be seen in AML [149]. In a meta-analysis, it was shown that *U2AF1* MDS not only had a worse OS but may have an increased AML transformation (HR 2.47, *p* = 0.0004) [150]. Bamopoulos et al. found *U2AF1* mutations in 3.4% of AML patients, which represented about 16.4% of splicing factor mutations (38/232) [151]. The data suggest that *U2AF1* mutations in AML may be associated with a poor OS [151,152]. The 2022 ELN risk stratification assigns *U2AF1* mutation in AML to the adverse risk category [21].

Given the frequency of spliceosome mutations in MDS, and the presence and poor prognosis of *U2AF1* mutations in AML, the development of therapeutic modalities to target this mutation are needed. In vitro and in vivo experiments suggest that *U2AF1*-mutated hematopoietic stem cells treated with splicesome modulation, via sudemycin D6, may have a decreased survival and proliferation [153]. Interleukin-1 receptor-associated kinases (IRAKs), which function in multiple inflammatory pathways, are suspected to play significant roles in hematologic malignancies such as AML and MDS [154]. It is thought that mutations in *U2AF1* and *SF3B1* create long isoforms of IRAK4 (IRAK4-long, or IRAK4-L), which are important in leukemogenesis; this creates a rationale for further research into the development of inhibitors of IRAK4, such as CA-4948 in AML [154,155,156]. A phase 1/2a study is recruiting patients to evaluate emavusertib, an IRAK-4 inhibitor, monotherapy in adults with AML or high-risk MDS, which includes patients with R/R AML or high-risk MDS with *U2AF1*, *SF3B1*, or *FLT3* mutations (NCT04278768) [157].

### 2.9. SF3B1

Mutations in the splicing factor 3B subunit 1A (*SF3B1*) are common in MDS [158]. In *SF3B1*-mutant MDS, several possible mechanistic explanations for disease have led to proposed disease targets [158], In MDS, Luspatercept has been shown to have clinical promise in low-risk MDS with ringed sideroblasts in the MEDALIST trial, and low-risk MDS in the COMMANDS trials, which led to FDA approval [159,160,161,162]. Importantly, secondary AML can develop from MDS, and one study identified mutations in RNA splicing (including *SF3B1*) as a common mutation seen in secondary AML [163]. The 2022 ELN risk stratification assigns an adverse risk to *SF3B1*-mutated AML [21]. However, data regarding the role of targeted treatment in *SF3B1*-mutant AML is lacking. A phase 1 clinical trial evaluated H3B-8800, which is a small molecule that binds to splicing factor 3B1 (*SF3B1*), in patients with MDS, CMML, and AML (38/84 patients had AML); there were no complete/partial responses by 2006 IWG criteria, and, per author, in the four patients with *SF3B1* mutations and AML, there were no RBC transfusion-independent periods [164]. As noted in the *U2AF1* section, there is a trial evaluating emavusertib (CA-4948) monotherapy in adults with R/R AML or high-risk MDS with the *SF3B1* mutation (NCT04278768), showing a promising early response [157].

### 2.10. SRSF2

Mutations in serine- and arginine-rich splicing factor 2 (SRSF2) are thought to play a role in MDS and may be associated with higher progression to AML [165,166,167]. *SRSF2* mutations tend to be associated with secondary AML [163]. In an analysis of the AMLCG-1999 trial, 151 patients aged greater than or equal to 75 years old were evaluated, and 38 patients (25%) had an *SRSF2* mutation [168]. However, other studies have demonstrated a lower frequency of *SRSF2* mutations, around 6–10% [151]. A study found that splicing factor mutations (including SF mutations other than *SRSF2*) commonly co-occur with *RUNX1, ASXL1, IDH2,* and *TET2* mutations, and not with *KIT, NPM1*, and *FLT3*-ITD mutations; of note, the co-occurrence of *SRSF2* mutations with other adverse risk mutations, such as RUNX1 or ASXL1, portends a lower EFS [169]. The 2022 ELN risk classification assigns *SRSF2* mutations to the adverse risk category [21]. Indeed, *SRSF2* gene mutations seem to portend a poor prognosis in patients with newly diagnosed AML treated with AZA-VEN [170]. Consolidation with allo-HSCT can benefit these patients in mitigating the adverse effect of *SRSF2* mutations, demonstrated by a 2-year OS of 77% in *SRSF2*-mutated AML patients after undergoing transplant; however, it should be noted that the continued detection of *SRSF2* mutations at HSCT (after induction of remission and before HSCT) does not seem to affect outcomes, and therefore may not be helpful as a predictive biomarker for treatment decisions [171]. There are currently no active clinical trials targeting *SRSF2* mutation.

### 2.11. C-KIT

*C-kit* is a proto-oncogene encoding a receptor tyrosine kinase which functions, upon binding to its ligand, in tyrosine residue phosphorylation and signal transduction; activating mutations of *c-kit* have been associated with neoplastic proliferation [172,173]. The c-kit receptor tyrosine kinase is believed to be involved in hematopoiesis and with an increased expression in AML [174]. Several studies have identified *c-kit* mutations in association with core binding factor acute myeloid leukemia (CBF AML) [175,176,177]. A systematic review of studies of CBF AML patients reported *c-kit* mutations in 10.9–46.2% of patients, with a mean of 31%; the authors concluded that *c-kit* mutations may indicate a poor prognosis in t(8;21) [176]. In an analysis of patients with inv(16)/t(16;16) or t(8;21)(q22;q22) CBF AML, 46.2% of patients had *c-kit* mutations; a lower OS was observed in t(8;21) AML patients with the *c-kit* TKD(816) mutation but not with c-kit mutations other than TKD(816) [175]. A study of 61 adult CBF AML patients with inv(16) and 49 adult CBF AML patients with t(8;21) showed a higher CIR of relapse (CIR) in mutated KIT for both inv(16) and t(8;21) [177].

The expression of c-kit (CD117) is thought to be present in about 60% of childhood and adult AML, with a higher expression associated with worse outcomes [178,179]. Though the impact of c-kit expression on outcomes may be controversial, one study found that c-kit expression measured by mean fluorescent index was associated with a poorer OS and PFS [180]. Imatinib mesylate has been studied as a maintenance therapy in patients with new *c-kit*-positive AML after induction and post-remission therapy; though the median PFS for younger patients ( < 60 years of age) was 52.1 months, which is longer relative to the historical controls, the study had many limitations, including a small sample size [178]. A phase 2 pilot study evaluated imatinib in *c-kit*-positive AML in patients who were not eligible for chemotherapy or who were chemo-refractory. Only 5 of 21 patients had a response [181]. Imatinib has also been studied in combination with induction chemotherapy in patients with relapsed AML in a phase 1 study, which showed a CR/CRp rate of 57% [182].

Nilotinib, a multikinase inhibitor that also targets the KIT tyrosine kinase, has been studied in combination with a 7 + 3 regimen in induction and in maintenance for AML patients with KIT expression [183]. In this phase II study, 61.7% of patients achieved CR/CRi [183]. The clinical trial NCT01830361 assessed the utility of the multi-kinase inhibitor midostaurin as an addition to standard intensive chemotherapy in patients with newly diagnosed AML with t(8;21) and mutations in *KIT* and/or *FLT3*-ITD; 16 patients had *KIT* mutations (88.9%), but the study did not reach the primary endpoint of an 80% 2-yr EFS [184]. Other studies have investigated imatinib alone or in combination with other medications in R/R settings (NCT01126814, NCT00707408). In CBF-AML, avapritinib is being studied in R/R or MRD-positive AML with *KIT* mutations (NCT05821738). A trial to evaluate the utility of the *c-KIT* mutation as an MRD marker in AML is ongoing (NCT06116318).

### 2.12. KRAS/NRAS

The *RAS* genes, which include *NRAS*, *KRAS*, and *HRAS*, are membrane-associated proteins involved in signal transduction [185]. In AML, *NRAS* mutations occur in about 11% of patients, while *KRAS* mutations occur in about 5% [185]. Other studies have reported higher occurrence of NRAS mutation of closer to 10–30% [17,186,187]. While the reported frequency of RAS mutations in AML vary, it seems that *NRAS* tends to occur more commonly in AML compared to *KRAS*/*HRAS* [186,188]. Generally, mutations in *RAS* are thought to occur in 10–25% of AML patients [189].

The impact of *RAS* mutations on AML prognosis remains unclear. Radich et al. did not find a statistically significant difference in CR or in survival in patients with *NRAS*-mutated AML compared to the unmutated group, and Bowen et al. did not find RAS mutations to affect OS, DFS, CR, or rate of relapse [185,186], In a study of 2502 patients with AML, 10.3% of patients had N-RAS mutations. The presence of *NRAS* mutations among patients with de novo AML, secondary AML, and therapy-related AML were similar, present in 10% of patients with de novo AML, 12.1% in patients with secondary AML, and 10.9% of patients with therapy-related AML [187]. Though Bacher et al. only report prognostic significance in normal karyotype, FLT3-unmutated, and *NRAS*-mutated AML (an improved EFS in the *NRAS*-mutant group compared to *NRAS* wild-type, *p* = 0.06) but otherwise report no significant prognostic association, Kiyoi et al. found that CR rates after initial induction were lower in the wild-type FLT3/mutant *NRAS* group (52%) than in the wild-type *FLT3*/wild-type NRAS group (79.7%), with *p* = 0.005 [17,187]. A Brazilian study did identify a shorter OS in *NRAS*-mutated AML patients compared to *NRAS*-unmutated AML patients [190].

A pre-clinical study provided a rationale for MEK inhibition in *NRAS*-mutated AML. *NRAS*-G12D AML-transplanted mice treated with MEK inhibition had an increased survival, though treatment did not affect cell death/differentiation [191]. MEK 1/2 inhibition has been studied in relapsed/refractory, RAS-positive myeloid malignancies in the phase 1/2 setting [192,193]. In a cohort of AML and MDS patients with a RAS mutation, treatment with trametinib in the relapsed/refractory setting yielded a 20% response rate with 8% CR (4/50 patients) [193]. Selumetinib, an oral MEK inhibitor, was studied in a phase 2 trial in patients with R/R AML or older untreated AML patients; in this study, 7% of patients had *NRAS* mutations and 2% of patients had *KRAS* mutations, but there was no response in the NRAS patients, and the only KRAS patient had an unconfirmed minor response with improvement in platelets [194]. A combination treatment strategy, given the known resistance that can occur with MEK inhibition, via AKT inhibition (PI3K/AKT pathway), in addition to MEK inhibition, has been studied in a phase 2 trial; the combination of GSK2141795 (pan-AKT kinase inhibitor) with MEK inhibition did not result in any complete remissions in a cohort of RAS-mutated AML patients [189].

## 3. Conclusions

The understanding of the ontology and biology of AML has dramatically progressed, with an emphasis on the impact of acquired somatic mutations and epigenetic changes on leukemogenesis. To varying degrees, the identification of particular genetic and epigenetic changes can have profound therapeutic implications. The identification of *FLT3-ITD*, *NPM1*, and *IDH1/2* have profoundly shifted diagnostic and therapeutic paradigms in AML, while data regarding the potential impact on prognosis and options for therapeutic targeting remain sparse for other somatic mutations. Adding further complexity to the molecular landscape of AML is the intricate interplay between various somatic mutations, germline mutations, and epigenetic changes, which in some cases can have a compounding influence on the prognosis.

Chemotherapy remains the backbone of treatment for AML, though targeted molecular treatments have been incorporated into induction, consolidation, and maintenance in extensively studied mutations such as *FLT3*-ITD. Table 2 highlights recent ongoing or completed trials for *TP53*, KMT2A, NPM1, KIT, *KRAS*/*NRAS*, *U2AF1*, *SF3B1*, *SRSF2*, *PHF6*, and *EZH2* mutations/alterations. Less is known about mutations such as *PHF6*, *EZH2*, and spliceosome mutations, highlighting the need for clinical trials.
biomedicines-12-01768-t002_Table 2Table 2Clinical trials for *TP53*, *KMT2A*, *NPM1*, *KIT*, *KRAS/NRAS*, *U2AF1*, *SF3B1*, *SRSF2*, *PHF6*, and *EZH2* mutations/alterations. Est = estimated. DLBCL = diffuse large B-cell lymphoma. CLL/SLL = chronic lymphocytic leukemia/small lymphocytic lymphoma. MM = multiple myeloma. ALL = acute lymphoblastic leukemia. MPAL = mixed-phenotype acute leukemia. ALAL = acute leukemias of ambiguous lineage. ITT= intent-to-treat analysis.MutationClinical Trial TreatmentPatient Cohort*n*Clinical PhaseStatusResponse RateReference*TP53*Nivolumab + decitabine + VENNew *TP53*-mutated AML1IActive, not recruiting

NCT04277442
*TP53*Entrectinib + ASTX727 (decitabine + cedazuridine)R/R AML with *TP53* mutation 12 (est)IRecruiting

NCT05396859
*TP53*Eprenetapopt (APR-246) + AZA (maintenance)In post-allogeneic HSCT setting, AML or MDS, with *TP53*
mutation33IICompletedmOS 20.6 months (AML and MDS)NCT03931291 [113]*TP53*Eprenetapopt (APR-246) + AZAMDS, MPN, MDS/MPN, CMML, non-proliferative AML, with *TP53* mutation52–53I/IIUnknownORR 33% in AMLNCT03588078 [112]*TP53*Eprenetapopt (APR-246) + AZAMDS, MDS/MPN, CMML, oligoblastic AML, with *TP53* mutation55I/IICompletedORR 64% and CR 36% in the AMLNCT03072043 [114]*KMT2A*, *NPM1*Revumenib (SNDX-5613) + induction/consolidation chemotherapy, and maintenance Newly diagnosed AML with *KMT2Ar*, *NPM1c*, or *NUP98r*
mutations76 (est)IRecruiting 
NCT06226571*KMT2A*, *NPM1*Revumenib (SNDX-5613) + chemotherapyR/R AML, ALL, ALAL with *KMT2Ar*, *KMT2A*
amplification, *NPM1c*, *NUP98r*30IActive, not recruiting 

NCT05326516
*KMT2A*, *NPM1*Revumenib (SNDX-5613) + “7 + 3” New AML with *NPM1* mutation or *KMT2Ar*28 (est)IRecruiting

NCT05886049
*KMT2A*, *NPM1*Revumenib (SNDX-5613) + cobicistat R/R acute leukemia (including ALL/MPAL or AML) including *NUP98r*, *NPM1* mutation, or *KMT2Ar*
(including pediatric population)413 (est)I/IIRecruiting

NCT04065399
*KMT2A*, *NPM1*Revumenib (SNDX-5613) + gilteritinib R/R AML with *FLT3* mutation and concurrent *MLL*
rearrangement or *NPM1*
mutation, or other mutations that cause HOXA-MEIS1 overexpression30 (est)IRecruiting

NCT06222580
*KMT2A*,*NPM1*Revumenib + venetoclaxAML with *KMT2Ar*, *NPM1*
mutation, or *NUP98r *(includes pediatric population)8 (est)I/IINot yet recruiting 
NCT06284486*KMT2A*, *NPM1*Ziftomenib + AZA + VEN, ziftomenib + VEN, ziftomenib + “7 + 3”Newly diagnosed or R/R AML with *KMT2Ar* or *NPM1*
mutation212 (est)IRecruiting

NCT05735184
*KMT2A*, *NPM1*Ziftomenib in combination depending on cohort R/R AML with *NPM1*
mutation or *KMT2Ar*, *FLT3* mutation171 (est)IRecruiting 

NCT06001788
*KMT2A*, *NPM1*Ziftomenib R/R AML with *NPM1*
mutation or *KMT2Ar*199 (est)I/IIRecruiting

NCT04067336
*KMT2A*, *NPM1*Ziftomenib in maintenance post-allogeneic HSCTAML post-allogeniec HSCT with *KMT2Ar* or *NPM1*
mutation22 (est)INot yet recruiting 

NCT06440135
*KMT2A*, *NPM1*JNJ-75276617 in combination with standard therapies depending on cohortDe novo AML, secondary AML, or R/R AML, with *KMT2A* or *NPM1*
alterations 150 (est)IRecruiting

NCT05453903
*KMT2A*, *NPM1*JNJ-75276617 with conventional chemotherapy Pediatric and young adult patients with R/R AML with *KMT2A *alterations, *NPM1 *alterations, or nucleoporin *(NUP98* or *NUP214)*alterations80 (est)INot yet recruiting 

NCT05521087
*KMT2A*, *NPM1*JNJ-75276617 (Bleximenib)R/R AML with *KMT2A-r*, *NPM1* mutations, *NUP98* alterations, or *NUP214* alterations (includes pediatric population)350 (est)I/IIRecruiting 

NCT04811560
*KMT2A*, *NPM1*DSP-5336R/R AML, ALL, or ALAL with *KMT2A* fusion or *NPM1* mutation70 (est)I/IIRecruiting
NCT04988555*KMT2A*, *NPM1*DS-1594b +/− chemotherapy (varying regimens)R/R AML or ALL, with *NPM1*
mutation or *MLLr*17I/IICompleted

NCT04752163
*KMT2A*Pinometostat (EPZ-5676)Advanced *MLL-r* leukemias 51ICompleted 1 patient with t(11;19) *MLL-r* AML had CRNCT01684150, [134]*KMT2A*Pinometostat (EPZ-5676) + azacitadine *MLL*rearrangement or partial tandem duplication translocation in AML (R/R or untreated patients deferring induction)1I/IICompleted 
NCT03701295 [135]*KIT*Avapritinib Core binding factor AML with *KIT *mutation (includes pediatric population)50 (est)IIRecruiting

NCT05821738
*KIT*Nilotinib + daunorubicin + cytarabine (“7 + 3”), then nilotinib + consolidation, then nilotinib maintenance New AML with *KIT*
expression and/or mutation34IICompletedCR/CRi 61.7% in ITT, CR/CRi 78% in evaluable patients NCT01806571 [183]*KIT*Imatinib (maintenance)Newly diagnosed *c-kit*-positive AML patients who have received induction and post-remission treatment 32IICompleted Median PFS 52.1 months (age < 60) and median PFS 10.7 months (age ≥ 60)NCT00509093 [178]*KRAS/NRAS*Cobimetinib + enasidenib mesylateR/R AML with *IDH2* mutation and *RAS*
pathway mutation 15 (est)IRecruiting 

NCT05441514
*U2AF1/SF3B1*Emavusertib (CA-4948)R/R AML with FLT3 mutations (previously treated with FLT3 inhibitor) or *U2AF1*/*SF3B1* mutation, or R/R high-risk MDS with *U2AF1*/*SF3B1* mutations366 (est)I/IIRecruiting

NCT04278768
*SRSF2*None





*PHF6*None





*EZH2*None

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
