# Peer review of "Molecular Features and Treatment Paradigms of Acute Myeloid Leukemia"

_biomedicines, 2024, doi:10.3390/biomedicines12081768_

Round 1
Reviewer 1 Report
Comments and Suggestions for Authors
This is comprehensive overview of molecular changes in AML with emphasis on treatment options available and new ongoing trials. The review is well presented and well written, and easy to follow although it is packed with data.
Author Response
This is comprehensive overview of molecular changes in AML with emphasis on treatment options available and new ongoing trials. The review is well presented and well written, and easy to follow although it is packed with data.
Thank you for your kind review
Reviewer 2 Report
Comments and Suggestions for Authors
The manuscript is very interesting and well-prepared . The authors describe the molecular features in AML - very important in diagnosis and prognosis and finally - in targeted treatment. The article presents actual knowledge about biology , somatic and germiline mutations in AML as well as the influence on therapeutic possibilities and results of clinical trials.
In my opinion, there is a lack of some historical information (e.g. in induction) concerning on the treatment of acute promyelocytic leukemia; the presence of the 15;17 translocation, which results the formation of the PML/RARA fusion protein and lack of promyelocyte differentiation .The therapies with the trans retinoic acid (ATRA) and arsenic trioxide (ATO) have changed the prognosis dramatically. Additionally, I propose to include the adverse effects observed in presented more advanced clinical trials . I have no other comments.
Author Response
In my opinion, there is a lack of some historical information (e.g. in induction) concerning on the treatment of acute promyelocytic leukemia; the presence of the 15;17 translocation, which results the formation of the PML/RARA fusion protein and lack of promyelocyte differentiation .The therapies with the trans retinoic acid (ATRA) and arsenic trioxide (ATO) have changed the prognosis dramatically. Additionally, I propose to include the adverse effects observed in presented more advanced clinical trials . I have no other comments.
Thank you for your response.
Due to limited amount of molecular changes that need to be covered in the review, we decided focus on non-APL leukemia and mutations that become more prognostically relevant in the past decade or so. The same constraint was the issue regarding the focus on the side effects and as we tried to provide efficacy as much as possible in many genes as we can. I hope answer satisfy your comment but we will be happy to add more if needed
Reviewer 3 Report
Comments and Suggestions for Authors
The review article entitled "Molecular features and treatment paradigms of acute myeloid leukemia" lacks significant novelty and does not substantially advance our understanding of the subject matter, which has been extensively covered by other works. The article is notably deficient in visual aids, containing no figures and only one table, which undermines its ability to effectively communicate complex information. Additionally, the references cited are predominantly older, with a marked absence of recent studies from 2024, which diminishes the article's relevance and currency. Given these shortcomings, the work does not meet the standards for publication, and I cannot endorse its acceptance in its current form.
Author Response
The review article entitled "Molecular features and treatment paradigms of acute myeloid leukemia" lacks significant novelty and does not substantially advance our understanding of the subject matter, which has been extensively covered by other works. The article is notably deficient in visual aids, containing no figures and only one table, which undermines its ability to effectively communicate complex information. Additionally, the references cited are predominantly older, with a marked absence of recent studies from 2024, which diminishes the article's relevance and currency. Given these shortcomings, the work does not meet the standards for publication, and I cannot endorse its acceptance in its current form.
Please see the attachment

Round 2
Reviewer 3 Report
Comments and Suggestions for Authors
The revised article titled "Molecular features and treatment paradigms of acute myeloid leukemia" shows substantial improvements compared to the initial version, particularly in its organization, depth of analysis, and clarity of presentation. The expanded sections on molecular targeting provide a comprehensive overview of the current state of AML research, making it more informative and relevant to the field. The inclusion of recent clinical trial data and a more detailed exploration of gene-specific therapies such as FLT3 and IDH1/2 mutations significantly enhances the article's value. However, some minor issues persist. Overall, the article is a commendable contribution to the literature on AML, offering valuable insights while indicating areas for further research.
1. Tables are not mentioned in the text. The manuscript includes two tables, none of which are mentioned in the main text. Strangely, Table 2 is presented after the conclusions.
Author Response
The revised article titled "Molecular features and treatment paradigms of acute myeloid leukemia" shows substantial improvements compared to the initial version, particularly in its organization, depth of analysis, and clarity of presentation. The expanded sections on molecular targeting provide a comprehensive overview of the current state of AML research, making it more informative and relevant to the field. The inclusion of recent clinical trial data and a more detailed exploration of gene-specific therapies such as FLT3 and IDH1/2 mutations significantly enhances the article's value. However, some minor issues persist. Overall, the article is a commendable contribution to the literature on AML, offering valuable insights while indicating areas for further research.
- Tables are not mentioned in the text. The manuscript includes two tables, none of which are mentioned in the main text. Strangely, Table 2 is presented after the conclusions.
Thank you for the comment. We have mentioned tables and the figure in the main text, as described below:
- Lines 191-193 – mention Table 1
- Lines 358-369 – mention Table 1
- Line 365 (Title of Table 1), removed “ITD” from FLT3-ITD, and instead placed “FLT3”
- Corrected spelling of emavusertib throughout paper
- Added details to patient population in Table 2 for NCT04278768
- Lines 689 – 691, added information about Table 2 in the conclusion
- Lines 693 – 694, added information to Table 2 description
- Line 710 – corrected misspelling of the word “critically”
- Defined ALL, ALAL and MPAL in table 2 description. Removed description of ALAL from NCT05326516 in KMT2A/NPM1